# Improving Recruitment for a Newborn Screening Pilot Study with Adaptations in Response to the COVID-19 Pandemic

**DOI:** 10.3390/ijns8020023

**Published:** 2022-03-22

**Authors:** Julia Wynn, Norma P. Tavakoli, Niki Armstrong, Jacqueline Gomez, Carrie Koval, Christina Lai, Stephanie Tang, Andrea Quevedo Prince, Yeyson Quevedo, Katrina Rufino, Laura Palacio Morales, Angela Pena, Sharon Grossman, Mary Monfiletto, Erika Ruda, Vania Jimenez, Lorraine Verdade, Ashley Jones, Michelle G. Barriga, Nandanee Karan, Alexandria Puma, Safa Sarker, Sarah Chin, Kelly Duarte, David H. Tegay, Irzaud Bacchus, Rajani Julooru, Breanne Maloney, Sunju Park, Akilan M. Saami, Lilian Cohen, Natasha Shapiro, Michele Caggana, Wendy K. Chung, Dorota Gruber

**Affiliations:** 1Department of Pediatrics, Columbia University Irving Medical Center, New York, NY 10032, USA; cl3981@cumc.columbia.edu (C.L.); stephanietang003@gmail.com (S.T.); akq2002@cumc.columbia.edu (A.Q.P.); yq2319@cumc.columbia.edu (Y.Q.); rufinokatiana@gmail.com (K.R.); lp2908@cumc.columbia.edu (L.P.M.); akilan.m@columbia.edu (A.M.S.); wck15@columbia.edu (W.K.C.); 2Wadsworth Center, New York State Department of Health, Division of Genetics, Albany, NY 12201, USA; norma.tavakoli@health.ny.gov (N.P.T.); breanne.maloney@health.ny.gov (B.M.); sunju.park@health.ny.gov (S.P.); michele.caggana@health.ny.gov (M.C.); 3Parent Project Muscular Dystrophy, Washington, DC 20005, USA; niki@parentprojectmd.org; 4NewYork-Presbyterian Hospital, New York, NY 10032, USA; jgomez@nyp.org (J.G.); anp9072@nyp.org (A.P.); shg9115@nyp.org (S.G.); 5NewYork-Presbyterian Lower Manhattan Hospital, New York, NY 10032, USA; 6NewYork-Presbyterian Weill Cornell Medical Center, New York, NY 10065, USA; mam9464@nyp.org; 7Department of Medicine, Columbia University Irving Medical Center, New York, NY 10032, USA; ck353@cumc.columbia.edu (C.K.); lil9022@med.cornell.edu (L.C.); 8NewYork-Presbyterian Queens, New York, NY 11432, USA; err9007@nyp.org (E.R.); vlj9002@nyp.org (V.J.); nas9132@nyp.org (N.S.); 9Department of Pediatrics, Cohen Children’s Medical Center, Northwell Health, New York, NY 11040, USA; lverdade@northwell.edu (L.V.); ajones29@northwell.edu (A.J.); mbarriga@northwell.edu (M.G.B.); nkaran@northwell.edu (N.K.); apuma4@northwell.edu (A.P.); ssarker@northwell.edu (S.S.); schin20@northwell.edu (S.C.); kduarte@northwell.edu (K.D.); dtegay@northwell.edu (D.H.T.); dgruber1@northwell.edu (D.G.); 10Feinstein Institutes for Medical Research, Health Information Exchange, Northwell Health, New York, NY 11040, USA; ibacchus@northwell.edu (I.B.); rjulooru@northwell.edu (R.J.); 11Departments of Pediatrics and Cardiology, Donald and Barbara Zucker School of Medicine at Hofstra/Northwell Health, New York, NY 11040, USA

**Keywords:** newborn screening, Duchenne Muscular Dystrophy, COVID-19 pandemic

## Abstract

Seven months after the launch of a pilot study to screen newborns for Duchenne Muscular Dystrophy (DMD) in New York State, New York City became an epicenter of the coronavirus disease 2019 (COVID-19) pandemic. All in-person research activities were suspended at the study enrollment institutions of Northwell Health and NewYork-Presbyterian Hospitals, and study recruitment was transitioned to 100% remote. Pre-pandemic, all recruitment was in-person with research staff visiting the postpartum patients 1–2 days after delivery to obtain consent. With the onset of pandemic, the multilingual research staff shifted to calling new mothers while they were in the hospital or shortly after discharge, and consent was collected via emailed e-consent links. With return of study staff to the hospitals, a hybrid approach was implemented with in-person recruitment for babies delivered during the weekdays and remote recruitment for babies delivered on weekends and holidays, a cohort not recruited pre-pandemic. There was a drop in the proportion of eligible babies enrolled with the transition to fully remote recruitment from 64% to 38%. In addition, the proportion of babies enrolled after being approached dropped from 91% to 55%. With hybrid recruitment, the proportion of eligible babies enrolled (70%) and approached babies enrolled (84%) returned to pre-pandemic levels. Our experience adapting our study during the COVID-19 pandemic led us to develop new recruitment strategies that we continue to utilize. The lessons learned from this pilot study can serve to help other research studies adapt novel and effective recruitment methods.

## 1. Introduction

Seven months after the launch of a pilot study to screen consented newborns for Duchenne Muscular Dystrophy (DMD) in New York State, New York City became an epicenter of the coronavirus disease 2019 (COVID-19) pandemic. While much was unknown in the initial weeks of the pandemic, it was apparent there was rapid spread of the virus and on 20 March 2020, the governor of New York State ordered NY pause, requiring all nonessential business to close. New York hospitals limited their clinical and research staff to only essential employees and in-person research recruitment that did not pose an immediate, significant benefit to the participant was halted, and all study staff were ordered to work from home [1]. The high rates of COVID-19 infection in labor and delivery units and early maternal deaths were particularly concerning. One study found that in late March and early April of 2020, approximately 15% of pregnant women admitted to two maternity units in northern Manhattan tested positive for SARS-CoV-2 [2].

The DMD newborn screening (NBS) pilot study is a collaboration of the Northwell Health and NewYork-Presbyterian (NYP) hospitals, New York State Department of Health (NYSDOH) Newborn Screening Program, the Newborn Screening Translational Research Network (NBSTRN) and is sponsored by Parent Project Muscular Dystrophy (PPMD). The goal of this pilot study is to gather feasibility data for NBS screening for DMD for nomination to the Recommended Uniform Screening Panel (RUSP). The first-tier screen for DMD is measurement of the concentration of creatine kinase-MM (CK-MM) in dried blood spot (DBS) specimens that are submitted for routine newborn screening using a previously described assay [3]. A pre-pilot study established cut-off CK-MM values for screen negative (no further testing required), borderline (a repeat DBS specimen requested for CK-MM screening), or screen positive (genetic counseling and recommendation for second-tier molecular testing for DMD and other neuromuscular conditions provided). A separate paper describes the pilot study in detail (manuscript in preparation).

The original recruitment method for the DMD pilot study and other NBS pilot studies we have conducted [4] relied on in-person recruitment of mothers in the postpartum unit approximately 24–48 h after delivery. This process ensures equity in recruitment and allows the NBS Program to complete the screening for the pilot study simultaneously with the routine NBS panel. Upon the mandate to suspend in-person research activities, there was a need for a sudden and immediate transition from 100% in-person recruitment to 100% remote recruitment. Additionally, it required the NBS Program to alter their DMD testing flow to accommodate participants who consented outside of the immediate newborn period. This transition was necessary for the health and safety of the patients, hospital employees and research staff.

Pilot studies are crucial to the success of research studies and assist in the addition of conditions to the U.S. recommended uniform screening panel (RUSP) [5,6]. Nomination of a condition to the RUSP requires prospective, population-based screening results which can be provided by a pilot study. Any disruption to a pilot study can negatively impact the research being performed and delay the groundwork for scientific research. To prevent the cessation of the Duchenne pilot study in NYS, we adapted the protocol to allow the continuation of the study during the COVID-19 pandemic.

We describe how the transition in recruitment was made and evolved. We conducted a post-hoc secondary analysis of the recruitment efforts of this pilot newborn screening study. The goal of this analysis was to investigate the effect of the changes in recruitment on the proportion eligible families approached to invite and enrolled as well as the proportion who declined participation and how these rates changed during the pandemic. Our experiences and the lessons learned have altered the way we conduct pilot NBS research and provided us with ways to minimize the impact of the future public health emergencies on the research studies affected by them and diversify existing recruitment methodology utilized during non-emergent states. We present our experience to guide other future research recruitment efforts.

## 2. Methods

### 2.1. Pre-Pandemic

Recruitment was conducted at the participating institutions of Northwell Health hospitals: Long Island Jewish (LIJ) Medical Center, North Shore University Hospital (NSUH), Lenox Hill Hospital (LHH), and Southside Hospital and NYP hospitals: NewYork-Presbyterian Hospital Morgan Stanley Children’s Hospital (MSCH); NewYork-Presbyterian Weill Cornell Medical Center (Cornell), Allen Hospital (Allen), Lower Manhattan Hospital (LMH) and NewYork-Presbyterian Queens (Queens). The electronic health records (EHR) were queried for mothers who had recently given birth. Exclusion criteria included mothers who did not speak English, Spanish, Mandarin or Cantonese. NYP hospitals also excluded babies who were in the neonatal intensive care unit for a structural birth defect or born prior to 34 weeks gestational age. At both institutions, some mothers under the age of 18 and those with documented psychiatric issues or religious observances were not approached per the discretion of the nursery staff. The study was approved by the Institutional Review Board (IRB) at all institutions participating in recruitment and at NSYDOH. Figure 1 provides a flow diagram of the recruitment and testing strategies for the three time periods.

At CHONY, Allen and Cornell enrollment started on 11 October 2019, LMH on 1 November 2019 and Queens on 1 February 2020. At Northwell Health hospitals, including LIJ, NSUH and LHH enrollment started on 8 October 2019 and at Southside Hospital on 19 February 2020. For the purposes of this analysis, we examined recruitment starting 1 January 2020.

Research assistants (RAs) entered each postpartum room during the first 72 h after delivery to approach parents to enroll their baby into the study for DMD screening. Parents had the opportunity to review the study brochure and a three-minute video about the study. Written consent was obtained from one parent or legal guardian by electronic consent within the REDCap© System [7] or by paper, according to the parent’s preference. All study materials were available in three languages and the RA team included native Spanish, Mandarin and Cantonese speaking individuals.

#### NYSDOH NBS Program Protocol

The RAs marked the dried blood spot (DBS) cards of consented babies before they were mailed to the NBS Program. Consent was also documented in the REDCap© System accessible to the NBS Program team. After the DBS specimens were received by the NBS Program, the consented specimens were separated from unconsented specimens and accessioned in the Laboratory Information Management System (LIMS) in consecutive order. Screening for the routine NBS panel and CK-MM analysis was conducted simultaneously.

The results of the CK-MM screen were reported through the routine NBS LIMS which disseminates results to the hospitals of birth and pediatricians. Borderline results, requiring repeat DBS specimen collection and submission, were communicated to the RA team by email, and the study team coordinated repeat DBS specimen collection and submission with the family and pediatrician. The results for normalized borderline results were reported by phone to the parent by the study RA or the pediatrician. Positive results were communicated to the study geneticist and genetic counselor, and the genetics team informed the pediatrician and contacted the family to schedule a study visit for genetic counseling and to arrange for molecular genetic testing, which required a new sample be collected by venous blood draw.

### 2.2. Remote Recruitment

#### 2.2.1. Participant Facing Protocol

In-person recruitment was halted on 13 March 2020 for the NYP and Northwell Health systems. In the week prior to this date, the study team took steps to prepare for remote recruitment including changes to the REDCap© System to add fields to collect contact information and track phone calls and emails to participants. A special COVID-19 protocol modification was submitted to the respective IRBs outlining remote recruitment of parents by phone and email. Remote recruitment was initiated on 16 March 2020 at NYP hospitals CHONY and LMH under a special COVID-19 circumstance to practice under pending modifications. Remote recruitment began at Cornell on 23 March 2020 after IRB approval at LMH on 20 April 2020. All research related activities were halted at Queens hospital during the fully remote recruitment period. Remote recruitment was initiated at all Northwell Health hospitals between 27–30 March 2020, except Southside Hospital where no recruitment was conducted during the fully remote recruitment period.

To minimize patients’ in-hospital exposure, early discharges were implemented in New York, which limited before discharge recruitment time. Screening eligibility was unchanged although babies of parents without a working phone number or email could not be recruited. At Northwell Health hospitals, when possible, parents were recruited remotely via phone during mothers’ and babies’ stay in the hospital or post-discharge when the mothers could not be contacted while in hospital. At NYP hospitals, all recruitment took place after discharge per the request of the postpartum units. Post-discharge recruitment usually took place within the first four weeks of life but after the routine NBS panel was completed. Recruitment was conducted by phone, or phone followed by email when email addresses were available. Up to three voicemails and emails were sent by the same RA in the preferred language identified in the electronic medical record system. A secure system was used to mask the RAs private cell phone number, and a study phone number and email were established to receive all incoming messages.

Parents who provided verbal consent by phone were emailed the consent in their preferred language and were given the option to stay on the line with the RA so they could assist with completion of the electronic consent. Parents who were not able to sign an electronic consent were mailed a paper consent with a stamped addressed return envelope and instructions to contact the RA if they had questions.

#### 2.2.2. NYSDOH NBS Program Protocol

As RAs were no longer on the postpartum floors, the DBS specimens were not marked before transport to the NBS Program. An additional step was implemented which included sending the NBS Program team a daily list of babies who had been consented to the study through a secure email. The list was crosschecked with the NBS LIMS to determine whether the specimen had been received by the NBS Program and if so, where it was in the testing process. As before the pandemic, consent was confirmed by the NBS Program staff through review of the REDCap© System.

When consent was obtained prior to the routine NBS panel testing initiation, as before COVID-19, the routine NBS panel and CK-MM screening were performed concurrently. More frequently, consent was not in the immediate neonatal period, and in these instances the routine NBS was completed and reported and then CK-MM screening was performed afterwards to prevent delays in reporting routine NBS results. After CK-MM screening, an amended result report was issued with the CK-MM screening result and an explanation for the amended report (i.e., late consent for a pilot study). Language on the report for borderline results was modified to request a repeat specimen “when practical” to reduce the burden on families and health care workers to return babies to a healthcare setting for repeat specimen collection.

To ensure that parents received screen negative results that were not issued at the time of the routine panel, parents and pediatricians were called or emailed a notification of normal results. Borderline and positive results were reported out in the same manner as pre-pandemic. The pre-pandemic protocol was followed for requested repeat samples for borderline results although, when possible, the repeat screen was collected at the same time as a scheduled pediatrician visit to try to limit risk of exposure. The screen positive protocol was changed to offer telehealth for the genetic counseling visit with the option to collect a buccal swab at home or visit a medical office for a venous blood draw. Parents were also counseled about the risks and benefits of delaying molecular testing given the COVID-19 threat at the time.

### 2.3. Hybrid In-Person and Remote Recruitment

#### 2.3.1. Participant Facing Protocol

When in-person research recruitment was again permitted by the hospital systems, a hybrid approach of pre-pandemic and remote recruitment was instituted. The IRB protocols at all sites were modified to allow for this hybrid approach. At the NYP hospitals, in-person recruitment resumed at CHONY on 1 August 2020, at Cornell and Allen and LMH on 1 October 2020 and at Queens on 1 December 2020. At Northwell Health hospitals, in-person recruitment resumed at LIJ, NSUH, and LHH on 13 July 2020 and at Southside on 20 August 2020. Remote recruitment was continued for babies who were discharged before RAs were able to approach them. We also began using remote recruitment for babies delivered on weekends, holidays, or days when the weather did not allow for the study team to travel to the hospital. As RAs were no longer waiting on site for mothers to be available to approach if they were initially unavailable, they used the additional time in their day to try to contact the families of all babies who were not approached during their hospital stay. Pre-pandemic no attempt was made to recruit these babies.

#### 2.3.2. NYSDOH NBS Program Protocol

The NBS staff used a combination of the two protocols described to handle specimens and perform laboratory testing. Northwell Health hospital systems returned to marking DBS specimens when possible. However, at NYP hospitals due to COVID restrictions marking of DBS specimens was limited by restrictions to access. Therefore, the majority of DBS specimens were unmarked when received by the NBS Program. The daily emailed list of consented babies and REDCap© system remained the primary source to confirm consent prior to testing. Similarly, accessioning and CK-MM screening, reporting results, and follow-up second-tier testing remained the same throughout the pandemic.

### 2.4. Calculation of Saturation, Enrollment and Decline Rates

The total number of eligible newborns was calculated for NYP hospitals and Northwell Health hospitals for each period of the study: pre-pandemic, remote, and hybrid. Eligible newborns is the total number of newborns who had NBS in a given time period excluding newborns who were not eligible for the study. On average, the percentage of noneligible newborns was approximately 15% of all babies at NYP and 4% at Northwell. When recruitment did not occur at a given hospital, the babies who had routine NBS at that hospital were excluded from the total number. The number of babies approached to participate in the study, enrolled in the study, and actively declined after being approached was tallied for each institution and period.

The approach saturation was calculated from the total approached as a percentage of the total newborns who were screened by the NBS program. The proportion enrolled/eligible is the total enrolled in the study as a percentage of the total newborns who were eligible for the study. The proportion enrolled/approached is the total enrolled as a percentage of the total approached, and proportion declined/approached is the total who actively declined after being approached as a percentage of total approached. The proportion passive declined/approached is the total who indicated consent to participate in the study but did not return a signed consent form. Enrollment rates were stratified by language of consent for each institution. Chi square analysis was used to compare proportions between pre-pandemic and the other periods of recruitment, remote, and hybrid. Analysis was completed in SAS [8].

For the NYP hospitals where these data were tracked, we also calculated the number of parents who provided verbal consent but did not complete the electronic consent, and the number of parents who were attempted to be recruited but could not be reached by phone or email.

## 3. Results

A total of 30,211 infants had newborn screening during the full time period of analysis and 22,664 were identified to be eligible for the DMD Pilot study. Pre-pandemic, 70% of the total eligible babies were approached and invited to the study, and 64% of those eligible were enrolled. Of those approached, 91% enrolled in the study and 9% actively declined to participate (Figure 2).

When the study was shifted to full remote recruitment, the proportion of families approached and enrolled fell to 68% (*p*-value < 0.00001) and the proportion enrolled of those eligible dropped to 38% (*p*-value < 0.00001). The proportion enrolled of those approached also fell to 55% (*p*-value < 0.00001) with 16% (*p*-value < 0.00001) actively declining and 29% (not a category pre-pandemic) passively declining by not completing a consent form after agreeing by phone (Figure 2). At NYP hospitals, an additional 713 (not included in the total approached tally) eligible families received an email message or voicemail inviting them to the study but did not reply. These data were not captured at Northwell Health hospitals.

After hybrid recruitment was initiated, 84% of eligible babies were approached which was an increase (*p*-value < 0.00001) from prior to the pandemic. The proportion of eligible babies who consented increased to 70% (*p*-value < 0.00001) for a total of 84% (*p*-value < 0.0001) of approached babies consenting to the study. The active decline proportion returned to a pre-pandemic level of 8% (*p*-value 0.6) and the passive decline proportion (8%) decreased compared to full remote recruitment as more recruitment was taking place in person (Figure 2). At NYP hospitals, an additional 452 (not included in the total approached tally) eligible families received an email message or voicemail inviting them to the study but did not reply.

### Language of Consent

The proportion of participants who completed consent in English, Spanish or Chinese remained relatively consistent across the periods of the study. English consents represented 92.3% during pre-pandemic, 88.9% during full remote, and 93.9% during hybrid. The same break down for Spanish was 5.9%, 9.4%, and 4.8% and for Chinese was 1.8%, 1.7%, and 1.3%, respectively.

## 4. Discussion

We describe the experiences of a multi-institutional NBS pilot study in New York City and Long Island, NY prior to the COVID-19 pandemic, at the start of the pandemic and several months into the pandemic. Ultimately, the modifications made to the study protocol resulted in more effective recruitment. Our experience and the lessons learned from the implementation of the modification to the protocol are particularly valuable for studies that rely on in-person recruitment. Incorporating remote recruitment may augment existing methods as well as allow for recruitment pivots when in-person recruitment is not possible.

Implementing remote recruitment during the pandemic allowed the study to recruit families to the study in a manner that protected the safety of the families and recruitment staff. After return to in person recruitment, a hybrid approach was pursued to allow the recruitment of families who delivered babies on weekends and holidays. These families were contacted by phone and email after discharge. The pivot to hybrid recruitment was possible by adapting our existing REDCap© System to track recruitment phone calls and emails and using the survey feature on REDCap© to send and obtain consents via email. Some challenges of remote recruitment were the lack of working contact information (phone or email) in the medical record, accurate notation of the patient’s preferred language to ensure the RA making contact spoke this language, and technology limitations of some families. In an attempt to overcome some of the technology challenges, RAs offered to assist the parent via prompts on the phone with the online consent. Families who did not have the ability to consent electronically were mailed a consent with study staff contact information and instructed to call to complete the paper consent over the phone.

Despite our efforts, when recruitment was fully remote a proportion of families who might have otherwise enrolled in the study did not enroll, likely in part due to inability of our research staff to contact them, and access and comfort with electronic consent. However, importantly, we did not observe a difference in the proportion of individuals consenting in the three study languages across the three time periods, indicating that one group was not disproportionately affected by barriers to enrollment.

Our study resources did not allow for other potential modes of contact or assistance that future studies may want to consider. Other studies have explored communicating via text with potential participants [9,10]. This enables individuals without email access or those who find texting more convenient to retain written documentation of their questions and responses. As use of patient portals increases, texting may also be a potential route of recruitment for pilot newborn screening studies as it allows a secure platform for confidential information.

The hybrid approach resulted in greater approach and enrollment saturation, illustrating the effectiveness of using remote recruitment to capture weekend and holiday deliveries that were normally missed if recruitment was limited to normal business days. The hybrid approach was achieved without increasing staff hours by re-configuring how time was spent. Prior to the hybrid approach, the RAs could not approach families after discharge from the hospital; therefore they spent additional time on the postpartum floors waiting for mothers to be available, as they were receiving medical care, nursing their baby, or sleeping. Frequently, mothers requested the RA to return at a more convenient time. With the hybrid approach, RAs did not wait for families who were not available and instead contacted them by phone. Thus, they had additional time to devote to remote recruitment. While enrollment rates did not return to pre-pandemic levels, this is reflective of individuals who consented but did not return the written consent and not due to an increase in active declines as these returned to pre-pandemic levels. Additional work is needed to refine the remote recruitment process to make it more accessible to improve consent completion rates. While an equivalent number of families were interested in participating, there remained barriers for the families who consented verbally but did not return the consent, potentially related to technology or simply the demands of being at home with a newborn.

## 5. Limitations

There are some limitations of this analysis of our pandemic recruitment experiences. While it is multi-institutional and multi-lingual, it is possible that some of our strategies are not generalizable to other studies. In particular, given that the parents were of reproductive age, they were potentially more adept with technology than other older populations. We used the total number of babies who had routine NBS as the total eligible population. Our data did not allow for a full analysis of if, or how, recruitment and enrollment differed by demographics of the families during each period. Previous literature indicates that certain populations may not be comfortable with remote recruitment and e-consent [11].

## 6. Conclusions

The COVID-19 pandemic in NYS in February and March 2020 led to discussions to consider pausing the pilot study unless alternative protocols could be rapidly established to allow the study to continue despite the restrictions caused by the pandemic. Instead, we made modifications to allow for successful remote recruitment. The new protocols could be applicable in the absence of a pandemic as complementary methods to in-person recruitment and follow-up, which would facilitate enrollment of patients who would otherwise be missed. Implementation of this hybrid approach optimizes patient enrollment and can be generalized.

## Figures and Tables

**Figure 1 IJNS-08-00023-f001:**
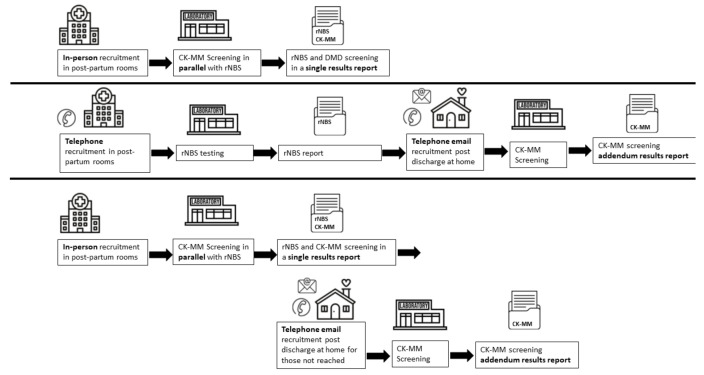
Recruitment, testing and reporting for the Duchene muscular dystrophy newborn screening. The three time periods of recruitment are depicted: pre-pandemic (**top** line), remote recruitment during the start of the pandemic (**middle** line), and hybrid recruitment (**bottom** line). During hybrid recruitment, phone and email recruitment were used to recruit babies who were delivered on weekends or holidays or were not available to recruit while they were in the hospital. Abbreviations: routine newborn screening (rNBS), creatine kinase-MM (CK-MM).

**Figure 2 IJNS-08-00023-f002:**
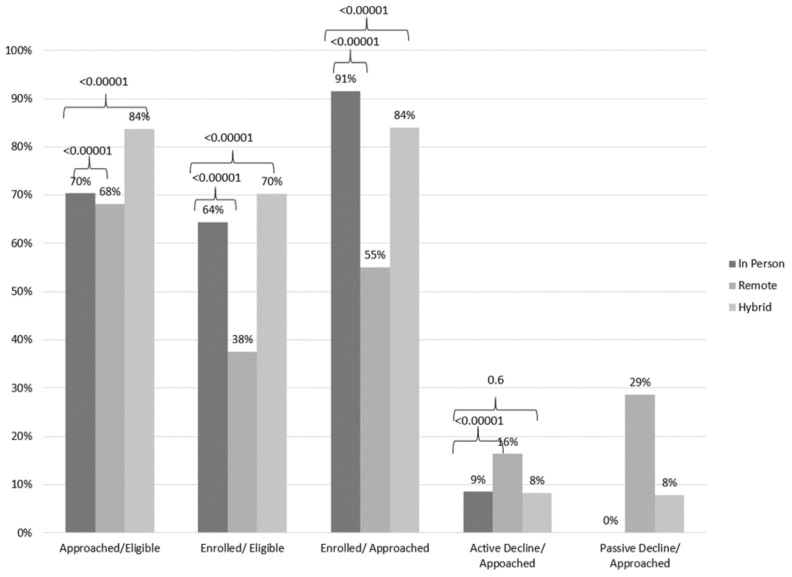
Approach and enrollment saturations and enrollment rates during pre-pandemic, fullremote and hybrid. Approach/Eligible is number approached as a fraction of total eligible babies with newborn screening (NBS), Enrolled/Eligible is number enrolled as a fraction of total eligible babies, Enrolled/Approached is number enrolled as a fraction of total approached, Active Decline/ Approached is the proportion who declined participation after being approached. Passive Decline/ Approached is the number who did not return a consent after being approached and agreeing to the study. The latter only exists for periods that had some remote recruitment. Recruitment took place at NewYork-Presbyterian (NYP) Hospitals: Children’s Hospital of New York, Allen, Cornell, and Lower Manhattan Hospital and Queens Hospital and Northwell Hospitals: Long Island Jewish Medical Center, North Shore University Hospital, Lenox Hill Hospital, and Southside Hospital unless otherwise indicated. Pre-pandemic period for NYP and Northwell includes 1 January 2020–15 March 2020. There was no recruitment at Queens between 1 January 2020–1 February 2020 and Southside for 1 January 2020–19 February 2020 and these hospitals are not included in total number for eligible babies for these time periods. The Remote period includes 16 Match 2020–1 September 2020 for NYP and 28 March 2020–12 July 2020 for Northwell. There was recruitment at Lower Manhattan Hospital from 16 Match 2020–20 April 2020 and no recruitment at NYP Queens or Southside for this total period and these hospitals are not included in total number eligible for these time periods. The Hybrid period includes 2 September 2020–31 December 2020 for NYP and 13 July 2020–31 December 2020 for Northwell. There was recruitment at NYP Queens from 2 September 2020–1 December 2020 and Southside from 13 July 2020–20 August 2020 and these hospitals are not included in total number for eligible babies for these dates. *p*-values for chi square analysis for pre-pandemic compared to remote or hybrid.

## Data Availability

The data presented in this study are available on request from the corresponding author.

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
