# Peer review of "Improving Recruitment for a Newborn Screening Pilot Study with Adaptations in Response to the COVID-19 Pandemic"

_2409-515X, 2022, doi:10.3390/ijns8020023_

Round 1

Reviewer 1 Report

The report is well written and easy to read.

It will be interesting when the pilot study is completed. Did the COVID-19 pandemic/shift in recruitment of babies have any impact on the coverage of babies being tracked and coming back for follow up after a positive screening result, false or true?

Reviewer 2 Report

No further comments

Reviewer 3 Report

The authors have responded adequately to the issues raised.

This manuscript is a resubmission of an earlier submission. The following is a list of the peer review reports and author responses from that submission.

Round 1

Reviewer 1 Report

Review: Improving recruitment for a newborn screening pilot study with adaptations in response to the COVID-19 pandemic, IJNS-1509302

 The MS describes the recruitment adaptions that were used through phases of the pandemic in a pilot DMD-NBS study. The main take home messages for me were that remote recruitment is not as effective as face-to-face recruitment (unsurprising) and that the improved recruitment was largely the result of capturing the week-end/holiday deliveries (not really COVID-related). Several points are noted for clarification/correction:

 For recruitment, it seems that invited and approached have the same meaning (L201) but both terms are used. Please use one or the other.

  1. In Results, L221, the results of 64% and 55% are different to what is shown in Fig 2
  2. L238: decreased should read a decrease
  3. L258: delete the
  4. L279: the hybrid recruitment seems the same as pre-pandemic except for the addition of remote recruitment of week-end/holiday deliveries. I dont understand how this was achieved without any increase in the time or number of employees. Please elaborate.
  5. Fig 1. Suggest adding a label in the bottom line to make it clear that telephone/email recruitment was for week-end/holiday deliveries
  6. Fig 2: the p-values for the Active Decline/Approached appear to be switched.

Reviewer 2 Report

The manuscript describes the challenges posed by COVID pandemic on an ongoing research. It serves as a model for researchers on how to continue a project, originally face-to-face and shifting to remote, then hybrid recruitment.

The creativity in implementing the project was evident with new ways of contacting patients and securing consent. The protocol needed revisions needing approval of IRB. This sends an excellent to the young researchers that every change in the protocol, even if valid, must be approved.

This hybrid format also introduced an opportunity for recruitment during off hours, weekends and holidays. This practice can be utilized even after the pandemic.

I have some minor concerns:

  1. page 6, line 258-259. The sentence is vague and needs to be improved.
  2. There were concerns on need for additional work to refine remote recruitment process. Can you explain this further, using lessons from this project?

Reviewer 3 Report

A well written report of the methods by which patients were recruited for a pilot study of neonatal screening for Duchenne Muscular Dystrophy before and during the Covid19 pandemic and how the consequent limitations in personal contact influenced the rates of families consenting to their baby to take part in the study.

The manuscript presented here describes the methodology for the recruitment of patients but is it really a scientific paper by itself?